# Efficient Minimax Signal Detection on Graphs

**Jing Qian**
Division of Systems Engineering
Boston University
Brookline, MA 02446
jingq@bu.edu

**Venkatesh Saligrama**
Department of Electrical and Computer Engineering
Boston University
Boston, MA 02215
srv@bu.edu

## Abstract

Several problems such as network intrusion, community detection, and disease outbreak can be described by observations attributed to nodes or edges of a graph. In these applications presence of intrusion, community or disease outbreak is characterized by novel observations on some unknown connected subgraph. These problems can be formulated in terms of optimization of suitable objectives on connected subgraphs, a problem which is generally computationally difficult. We overcome the combinatorics of connectivity by embedding connected subgraphs into linear matrix inequalities (LMI). Computationally efficient tests are then realized by optimizing convex objective functions subject to these LMI constraints. We prove, by means of a novel Euclidean embedding argument, that our tests are minimax optimal for exponential family of distributions on 1-D and 2-D lattices. We show that internal conductance of the connected subgraph family plays a fundamental role in characterizing detectability.

## 1   Introduction

Signals associated with nodes or edges of a graph arise in a number of applications including sensor network intrusion, disease outbreak detection and virus detection in communication networks. Many problems in these applications can be framed from the perspective of hypothesis testing between null and alternative hypothesis. Observations under null and alternative follow different distributions. The alternative is actually composite and identified by sub-collections of connected subgraphs.

To motivate the setup consider the disease outbreak problem described in [1]. Nodes there are associated with counties and observations associated with each county correspond to reported cases of a disease. Under the null distribution, observations at each county are assumed to be poisson distributed and independent across different counties. Under the alternative there are a contiguous sub-collection of counties (connected sub-graph) that each experience elevated cases on average from their normal levels but are otherwise assumed to be independent. The eventual shape of the sub-collection of contiguous counties is highly unpredictable due to uncontrollable factors.

In this paper we develop a novel approach for signal detection on graphs that is both statistically effective and computationally efficient. Our approach is based on optimizing an objective function subject to subgraph connectivity constraints, which is related to generalized likelihood ratio tests (GLRT). GLRTs maximize likelihood functions over combinatorially many connected subgraphs, which is computationally intractable. On the other hand statistically, GLRTs have been shown to be asymptotically minimax optimal for exponential class of distributions on Lattice graphs & Trees [2] thus motivating our approach.We deal with combinatorial connectivity constraints by obtaining a novel characterization of connected subgraphs in terms of convex Linear Matrix Inequalities (LMIs). In addition we show how our LMI constraints naturally incorporate other features such as shape and size. We show that the resulting tests are essentially minimax optimal for exponential family

of distributions on 1-D and 2-D lattices. Conductance of the subgraph, a parameter in our LMI constraint, plays a central role in characterizing detectability.

**Related Work:** The literature on signal detection on graphs can be organized into parametric and non-parametric methods, which can be further sub-divided into computational and statistical analysis themes. Parametric methods originated in the scan statistics literature [3] with more recent work including that of [4, 5, 6, 1, 7, 8] focusing on graphs. Much of this literature develops scanning methods that optimize over rectangles, circles or neighborhood balls [5, 6] across different regions of the graphs. However, the drawbacks of simple shapes and the need for non-parametric methods to improve detection power is well recognized. This has led to new approaches such as simulated annealing [5, 4] but is lacking in statistical analysis. More recent work in ML literature [9] describes semi-definite programming algorithm for non-parametric shape detection, which is similar to our work here. However, unlike us their method requires a heuristic rounding step, which does not lend itself to statistical analysis. In this context a number of recent papers have focused on statistical analysis [10, 2, 11, 12] with non-parametric shapes. They derive fundamental bounds for signal detection for the elevated means testing problem in the Gaussian setting on special graphs such as trees and lattices. In this setting under the null hypothesis the observations are assumed to be independent identically distributed (IID) with standard normal random variables. Under the alternative the Gaussian random variables are assumed to be standard normal except on some *connected subgraph* where the mean $\mu$ is elevated. They show that GLRT achieves "near"-minimax optimality in a number of interesting scenarios. While this work is interesting the suggested algorithms are computationally intractable. To the best of our knowledge only [13, 14] explores a computationally tractable approach and also provides statistical guarantees. Nevertheless, this line of work does not explicitly deal with connected subgraphs (complex shapes) but deals with more general clusters. These are graph partitions with small out-degree. Although this appears to be a natural relaxation of connected subgraphs/complex-shapes it turns out to be quite loose[1] and leads to substantial gap in statistical effectiveness for our problem. In contrast we develop a new method for signal detection of complex shapes that is not only statistically effective but also computationally efficient.

## 2 Problem Formulation

Let $G = (V, E)$ denote an undirected unweighted graph with $|V| = n$ nodes and $|E| = m$ edges. Associated with each node, $v \in V$, are observations $x_v \in \mathbb{R}^p$. We assume observations are distributed $\mathbb{P}_0$ under the null hypothesis. The alternative is composite and the observed distribution, $\mathbb{P}_S$, is parameterized by $S \subseteq V$ belonging to a class of subsets $\Lambda \subseteq \mathcal{S}$, where $\mathcal{S}$ is the superset. We denote by $\mathcal{S}_K \subseteq \mathcal{S}$ the collection of size-$K$ subsets. $E_S = \{(u, v) \in E : u \in S, v \in S\}$ denotes the induced edge set on $S$. We let $x_S$ denote the collection of random variables on the subset $S \subseteq V$. $S^c$ denotes nodes $V - S$. Our goal is to design a decision rule, $\pi$, that maps observations $x^n = (x_v)_{v \in V}$ to $\{0, 1\}$ with zero denoting null hypothesis and one denoting the alternative. We formulate risk following the lines of [12] and combine Type I and Type II errors:

$$R(\pi) \quad = \quad \mathbb{P}_0 \left( \pi(x^n) = 1 \right) + \max_{S \in \Lambda} \mathbb{P}_S \left( \pi(x^n) = 0 \right) \tag{1}$$

**Definition 1** ($\delta$-Separable). We say that the composite hypothesis problem is $\delta$-separable if there exists a test $\pi$ such that, $R(\pi) \leq \delta$.

We next describe asymptotic notions of detectability and separability. These notions requires us to consider large-graph limits. To this end we index a sequence of graphs $G_n = (V_n, E_n)$ with $n \to \infty$ and an associated sequence of tests $\pi_n$.

**Definition 2** (Separability). We say that the composite hypothesis problem is asymptotically $\delta$-separable if there is some sequence of tests, $\pi_n$, such that $R(\pi_n) \leq \delta$ for sufficiently large $n$. It is said to be asymptotically separable if $R(\pi_n) \longrightarrow 0$. The composite hypothesis problem is said to be asymptotically inseparable if no such test exists.

Sometimes, additional granular measures of performance are often useful to determine asymptotic behavior of Type I and Type II error. This motivates the following definition:

**Definition 3** ($\delta$-Detectability). We say that the composite hypothesis testing problem is $\delta$-detectable if there is a sequence of tests, $\pi_n$, such that,

$$\sup_{S \in \Lambda} \mathbb{P}_S(\pi_n(x^n) = 0) \overset{n \to \infty}{\longrightarrow} 0, \quad \limsup_n \mathbb{P}_0(\pi_n(x^n) = 1) \leq \delta$$

In general $\delta$-detectability does not imply separability. For instance, consider $x \overset{H_0}{\sim} \mathcal{N}(0, \sigma^2)$ and $x \overset{H_1}{\sim} \mathcal{N}(\mu, \frac{\sigma^2}{n})$. It is $\delta$-detectable for $\frac{\mu}{\sigma} \geq 2\sqrt{\log \frac{1}{\delta}}$ but not separable.

**Generalized Likelihood Ratio Test (GLRT)**   is often used as a statistical test for composite hypothesis testing. Suppose $\phi_0(x^n)$ and $\phi_S(x^n)$ are probability density functions associated with $\mathbb{P}_0$ and $\mathbb{P}_S$ respectively. The GLRT test thresholds the "best-case" likelihood ratio, namely,

$$\text{GLRT:} \quad \ell_{\max}(x^n) = \max_{S \in \Lambda} \ell_S(x^n) \underset{H_0}{\overset{H_1}{\gtrless}} \eta, \quad \ell_S(x) = \log \frac{\phi_S(x^n)}{\phi_0(x^n)} \tag{2}$$

*Local Behavior*: Without additional structure, the likelihood ratio, $\ell_S(x)$ for a fixed $S \in \Lambda$ is a function of observations across all nodes. Many applications exhibit *local behavior*, namely, the observations under the two hypothesis behave distinctly only on some small subset of nodes (as in disease outbreaks). This justifies introducing local statistical models in the following section. *Combinatorial*: The class $\Lambda$ is combinatorial such as collections of connected subgraphs and GLRT is not generally *computationally* tractable. On the other hand GLRT is minimax optimal for special classes of distributions and graphs and motivates development of tractable algorithms.

## 2.1  Statistical Models & Subgraph Classes

The foregoing discussion motivates introducing local models, which we present next. Then informed by existing results on separability we categorize subgraph classes by shape, size and connectivity.

### 2.1.1  Local Statistical Models

*Signal in Noise Models* arise in sensor network (SNET) intrusion [7, 15] and disease outbreak detection [1]. They are modeled with Gaussian (SNET) and Poisson (disease outbreak) distributions.

$$\mathbf{H_0}: \ x_v = w_v; \quad \mathbf{H_1}: \ x_v = \mu \alpha_{uv} \mathbf{1}_S(v) + w_v, \ \text{for some}, \ S \in \Lambda, \ u \in S \tag{3}$$

For Gaussian case we model $\mu$ as a constant, $w_v$ as IID standard normal variables, $\alpha_{uv}$ as the propagation loss from source node $u \in S$ to the node $v$. In disease outbreak detection $\mu = 1$, $\alpha_{uv} \sim Pois(\lambda N_v)$ and $w_v \sim Pois(N_v)$ are independent Poisson random variables, and $N_v$ is the population of county $v$. In these cases $\ell_S(x)$ takes the following local form where $Z_v$ is a normalizing constant.

$$\ell_S(x) = \ell_S(x_S) \propto \sum_{v \in V} (\Psi_v(x_v) - \log(Z_v)) \mathbf{1}_S(v) \tag{4}$$

We characterize $\mu_0, \lambda_0$ as the minimum value that ensures separability for the different models:

$$\mu_0 = \inf\{\mu \in \mathbb{R}^+ \mid \exists \pi_n, \lim_{n \to \infty} R(\pi_n) = 0\}, \ \lambda_0 = \inf\{\lambda \in \mathbb{R}^+ \mid \exists \pi_n, \lim_{n \to \infty} R(\pi_n) = 0\} \tag{5}$$

*Correlated Models* arise in textured object detection [16] and protein subnetwork detection [17]. For instance consider a common random signal $z$ on $S$, which results in uniform correlation $\rho > 0$ on $S$.

$$\mathbf{H_0}: \ x_v = w_v; \quad \mathbf{H_1}: \ x_v = (\sqrt{\rho(1-\rho)^{-1}})z\mathbf{1}_S(v) + w_v, \ \text{for some}, \ S \in \Lambda, \tag{6}$$

$z, w_v$ are standard IID normal random variables. Again we obtain $\ell_S(x) = \ell_S(x_S)$. These examples motivate the following general setup for local behavior:

**Definition 4.** The distributions $\mathbb{P}_0$ and $\mathbb{P}_S$ are said to exhibit *local structure* if they satisfy:
**(1) Markovianity**: The null distribution $\mathbb{P}_0$ satisfies the properties of a Markov Random Field (M-RF). Under the distribution $\mathbb{P}_S$ the observations $x_S$ are conditionally independent of $x_{S_1^c}$ when conditioned on annulus $S_1 \cap S^c$, where $S_1 = \{v \in V \mid d(v, w) \leq 1, \ w \in S\}$, is the 1-neighborhood of $S$. **(2) Mask**: Marginal distributions of observations under $\mathbb{P}_0$ and $\mathbb{P}_S$ on nodes in $S^c$ are identical: $\mathbb{P}_0(x_{S^c} \in A) = \mathbb{P}_S(x_{S^c} \in A), \ \forall A \in \mathcal{A}$, the $\sigma$-algebra of measurable sets.
**Lemma 1** ([7])**.** *Under conditions (1) and (2) it follows that $\ell_S(x) = \ell_S(x_{S_1})$.*

### 2.1.2 Structured Subgraphs

Existing works [10, 2, 12] point to the important role of size, shape and connectivity in determining detectability. For concreteness we consider the signal in noise model for Gaussian distribution and tabulate upper bounds from existing results for $\mu_0$ (Eq. 5). The lower bounds are messier and differ by logarithmic factors but this suffices for our discussion here. The table reveals several important points. Larger sets are easier to detect – $\mu_0$ decreases with size; connected $K$-sets are easier to detect relative to arbitrary $K$-sets; for 2-D lattices "thick" connected shapes are easier to detect than "thin" sets (paths); finally detectability on complete graphs is equivalent to arbitrary $K$-sets, i.e., shape does not matter. Intuitively, these tradeoffs make sense. For a constant $\mu$, "signal-to-noise" ratio increases with size. Combinatorially, there are fewer $K$-connected sets than arbitrary $K$-sets; fewer connected balls than connected paths; and fewer connected sets in 2-D lattices than dense graphs. These results point to the need for characterizing the signal detection problem in terms of

|  | Arbitrary $K$-Set | $K$-Connected Ball | $K$-Connected Path |
|---|---|---|---|
| Line Graph | $\omega\left(\sqrt{2\log(n)}\right)$ | $\omega\left(\sqrt{\frac{2}{K}\log(n)}\right)$ | $\omega\left(\sqrt{\frac{2}{K}\log(n)}\right)$ |
| 2-D Lattice | $\omega\left(\sqrt{2\log(n)}\right)$ | $\omega\left(\sqrt{\frac{2}{K}\log(n)}\right)$ | $\omega\left(1\right)$ |
| Complete | $\omega\left(\sqrt{2\log(n)}\right)$ | $\omega\left(\sqrt{2\log(n)}\right)$ | $\omega\left(\sqrt{2\log(n)}\right)$ |

connectivity, size, shape and the properties of the ambient graph. We also observe that the table is somewhat incomplete. While balls can be viewed as thick shapes and paths as thin shapes, there are a plethora of intermediate shapes. A similar issue arises for sparse vs. dense graphs. We introduce general definitions to categorize shape and graph structures below.

**Definition 5** (Internal Conductance). (a.k.a. Cut Ratio) Let $H = (S, F_S)$ denote a subgraph of $G = (V, E)$ where $S \subseteq V$, $F_S \subseteq E_S$, written as $H \subseteq G$. Define the internal conductance of $H$ as:

$$\phi(H) = \min_{A \subset S} \frac{|\delta_S(A)|}{\min\{|A|, |S - A|\}}; \quad \delta_S(A) = \{(u, v) \in F_S \mid u \in A,\ v \in S - A\} \tag{7}$$

Apparently $\phi(H) = 0$ if $H$ is not connected. The internal conductance of a collection of subgraphs, $\Sigma$, is defined as the smallest internal conductance:

$$\phi(\Sigma) = \min_{H \in \Sigma} \phi(H)$$

For future reference we denote the collection of connected subgraphs by $\mathcal{C}$ and by $\mathcal{C}_{a,\Phi}$ the sub-collections containing node $a \in V$ with minimal internal conductance $\Phi$:

$$\mathcal{C} = \{H \subseteq G : \phi(H) > 0\}, \quad \mathcal{C}_{a,\Phi} = \{H = (S, F_S) \subseteq G : a \in S, \phi(H) \geq \Phi\} \tag{8}$$

In 2-D lattices, for example, $\phi(B_K) \approx \Omega(1/\sqrt{K})$ for connected K-balls $B_K$ or other thick shapes of size $K$. $\phi(\mathcal{C} \cap \mathcal{S}_K) \approx \Omega(1/K)$ due to "snake"-like thin shapes. Thus internal conductance explicitly accounts for shape of the sets.

## 3 Convex Programming

We develop a convex optimization framework for generating test statistics for local statistical models described in Section 2.1. Our approach relaxes the combinatorial constraints and the functional objectives of the GLRT problem of Eq.(2). In the following section we develop a new characterization based on linear matrix inequalities that accounts for size, shape and connectivity of subgraphs.

For future reference we denote $A \circ B \triangleq [A_{ij} B_{ij}]_{i,j}$.

Our first step is to embed subgraphs, $H$ of $G$, into matrices. A binary symmetric incidence matrix, $A$, is associated with an undirected graph $G = (V, E)$, and encodes edge relationships. Formally, the edge set $E$ is the support of $A$, namely, $E = \text{Supp(A)}$. For subgraph correspondences we consider symmetric matrices, $M$, with components taking values in the unit interval, $[0, 1]$.

$$\mathcal{M} = \{M \in [0, 1]^{n \times n} \mid M_{uv} \leq M_{uu},\ M\ \text{Symmetric}\}$$

**Definition 6.** $M \in \mathcal{M}$ is said to correspond to a subgraph $H = (S, F_S)$, written as $H \rightleftharpoons M$, if
$$S = \text{Supp}\{\text{Diag}(M)\}, \ F_S = \text{Supp}(A \circ M)$$

The role of $M \in \mathcal{M}$ is to ensure that if $u \notin S$ we want the corresponding edges $M_{uv} = 0$. Note that $A \circ M$ in Defn. 6 removes the spurious edges $M_{uv} \neq 0$ for $(u, v) \notin E_S$.

Our second step is to characterize connected subgraphs as convex subsets of $\mathcal{M}$. Now a subgraph $H = (S, F_S)$ is a connected subgraph if for every $u, v \in S$, there is a path consisting only of edges in $F_S$ going from $u$ to $v$. This implies that for two subgraphs $H_1$, $H_2$ and corresponding matrices $M_1$ and $M_2$, their convex combination $M_\eta = \eta M_1 + (1 - \eta)M_2$, $\eta \in (0, 1)$ naturally corresponds to $H = H_1 \cup H_2$ in the sense of Defn 6. On the other hand if $H_1 \cap H_2 = \emptyset$ then $H$ is disconnected and so $M_\eta$ is as well. This motivates our convex characterization with a common "anchor" node. To this end we consider the following collection of matrices:
$$\mathcal{M}_a^* = \{M \in \mathcal{M} \mid M_{aa} = 1, M_{vv} \leq M_{av}\}$$
Note that $\mathcal{M}_a^*$ includes star graphs induced on subsets $S = \text{Supp}(\text{Diag}(M))$ with anchor node $a$. We now make use of the well known properties [18] of the Laplacian of a graph to characterize connectivity. The unnormalized Laplacian matrix of an undirected graph $G$ with incidence matrix $A$ is described by $L(A) = \text{diag}(A\mathbf{1}_n) - A$ where $\mathbf{1}_n$ is the all-one vector.

**Lemma 2.** *Graph $G$ is connected if and only if the number of zero eigenvalues of $L(A)$ is one.*

Unfortunately, we cannot directly use this fact on the subgraph $A \circ M$ because there are many zero eigenvalues because the complement of $\text{Supp}(\text{Diag}(M))$ is by definition zero. We employ linear matrix inequalities (LMI) to deal with this issue. The condition [19] $F(x) = F_0 + F_1 x_1 + \cdots + F_p x_p \succeq 0$ with symmetric matrices $F_j$ is called a linear matrix inequality in $x_j \in \mathbb{R}$ with respect to the positive semi-definite cone represented by $\succeq$. Note that the Laplacian of the subgraph $L(A \circ M)$ is a linear matrix function of $M$. We denote a collection of subgraphs as follows:
$$\mathcal{C}_{LMI}(a, \gamma) \triangleq \{H \rightleftharpoons M \mid M \in \mathcal{M}_a^*, \ L(A \circ M) - \gamma L(M) \succeq 0\} \qquad (9)$$

**Theorem 3.** *The class $\mathcal{C}_{LMI}(a, \gamma)$ is connected for $\gamma > 0$. Furthermore, every connected subgraph can be characterized in this way for some $a \in V$ and $\gamma > 0$, namely, $\mathcal{C} = \bigcup_{a \in V, \gamma > 0} \mathcal{C}_{LMI}(a, \gamma)$.*

*Proof Sketch.* $M \in \mathcal{C}_{LMI}(a, \gamma)$ implies $M$ is connected. By definition of $\mathcal{M}_a$ there must be a star graph that is a subgraph on $\text{Supp}(\text{Diag}(M))$. This means that $L(M)$ (hence $L(A \circ M)$) can only have one zero eigenvalue on $\text{Supp}(\text{Diag}(M))$. We can now invoke Lemma 2 on $\text{Supp}(\text{Diag}(M))$. The other direction is based on hyperplane separation of convex sets. Note that $\mathcal{C}_{a,\gamma}$ is convex but $\mathcal{C}$ is not. This necessitates the need for an anchor. In practice this means that we have to search for connected sets with different anchors. This is similar to scan statistics the difference being that we can now optimize over arbitrary shapes. We next get a handle on $\gamma$.

$\gamma$ **encodes Shape:** We will relate $\gamma$ to the internal conductance of the class $\mathcal{C}$. This provides us with a tool to choose $\gamma$ to reflect the type of connected sets that we expect for our alternative hypothesis. In particular thick sets correspond to relatively large $\gamma$ and thin sets to small $\gamma$. In general for graphs of fixed size the minimum internal conductance over all connected shapes is strictly positive and we can set $\gamma$ to be this value if we do not a priori know the shape.

**Theorem 4.** *In a 2-D lattice, it follows that $\mathcal{C}_{a,\Phi} \subseteq \mathcal{C}_{LMI}(a, \gamma)$, where $\gamma = \Theta(\frac{\Phi^2}{\log(1/\Phi)})$.*

**LMI-Test:** We are now ready to present our test statistics. We replace indicator variables with the corresponding matrix components in Eq. 4, i.e., $\mathbf{1}_S(v) \rightarrow M_{vv}$, $\mathbf{1}_S(u)\mathbf{1}_S(v) \rightarrow M_{uv}$ and obtain:

Elevated Mean: $\qquad \ell_M(x) = \sum_{v \in V} (\Psi_v(x_v) - \log(Z_v))M_{vv}$

Correlated Gaussian: $\quad \ell_M(x) \propto \sum_{(u,v) \in E} \Psi(x_u, x_v)M_{uv} - \sum_v M_{vv} \log(1 - \rho) \qquad (10)$

$$\text{LMIT}_{a,\gamma} \qquad \qquad \ell_{a,\gamma}(x) = \max_{M \in \mathcal{C}_{LMI}(a,\gamma)} \ell_M(x) \overset{H_1}{\underset{H_0}{\gtrless}} \eta \qquad (11)$$

This test explicitly makes use of the fact that alternative hypothesis is anchored at $a$ and the internal conductance parameter $\gamma$ is known. We will refine this test to deal with the completely agnostic case in the following section.

# 4 Analysis

In this section we analyze $\text{LMIT}_{a,\gamma}$ and the agnostic LMI tests for the Elevated Mean problem for exponential family of distributions on 2-D lattices. For concreteness we focus on Gaussian & Poisson models and derive lower and upper bounds for $\mu_0$ (see Eq. 5). Our main result states that to guarantee separability, $\mu_0 \approx \Omega\left(\frac{1}{K\Phi}\right)$, where $\Phi$ is the internal conductance of the family $\mathcal{C}_{a,\Phi}$ of connected subgraphs, $K$ is the size of the subgraphs in the family, and $a$ is some node that is common to all the subgraphs. The reason for our focus on homogenous Gaussian/Poisson setting is that we can extend current lower bounds in the literature to our more general setting and demonstrate that they match the bounds obtained from our LMIT analysis. We comment on how our LMIT analysis extends to other general structures and models later.

The proof for LMIT analysis involves two steps (see Supplementary):

1. *Lower Bound:* Under $H_1$ we show that the ground truth is a feasible solution. This allows us to lower bound the objective value, $\ell_{a,\gamma}(x)$, of Eq. 11.
2. *Upper Bound:* Under $H_0$ we consider the dual problem. By weak duality it follows that any feasible solution of the dual is an upper bound for $\ell_{a,\gamma}(x)$. A dual feasible solution is then constructed through a novel Euclidean embedding argument.

We then compare the upper and lower bounds to obtain the critical value $\mu_0$.

We analyze both non-agnostic and agnostic LMI tests for the homogenous version of Gaussian and Poisson models of Eq. 3 for both finite and asymptotic 2-D lattice graphs. For the finite case the family of subgraphs in Eq. 3 is assumed to belong to the connected family of sets, $\mathcal{C}_{a,\Phi} \cap \mathcal{S}_K$, containing a fixed common node $a \in V$ of size $K$. For the asymptotic case we let the size of the graph approach infinity ($n \to \infty$). For this case we consider a sequence of connected family of sets $\mathcal{C}^n_{a.\Phi_n} \cap \mathcal{S}_{K_n}$ on graph $G_n = (V_n, E_n)$ with some fixed anchor node $a \in V_n$. We will then describe results for agnostic LMI tests, i.e., lacking knowledge of conductance $\Phi$ and anchor node $a$.

**Poisson Model:** In Eq. 3 we let the population $N_v$ to be identically equal to one across counties. We present LMI tests that are *agnostic* to shape and anchor nodes:

$$\text{LMIT}_A : \quad \ell(x) = \max_{a \in V, \gamma \geq \Phi^2_{min}} \sqrt{\gamma} \ell_{a,\gamma}(x) \overset{H_0}{\underset{H_1}{\gtrless}} 0 \tag{12}$$

where $\Phi_{min}$ denotes the minimum possible conductance of a connected subgraph with size $K$, which is $2/K$.

**Theorem 5.** *The $LMIT_{a,\gamma}$ test achieves $\delta$-separability for $\lambda = \Omega(\frac{\log(K)}{K\Phi})$ and the agnostic test $LMIT_A$ for $\lambda = \Omega(\log K \sqrt{\log n})$.*

Next we consider the asymptotic case and characterize tight bounds for separability.

**Theorem 6.** *The two hypothesis $H_0$ and $H_1$ are asymptotically inseparable if $\lambda_n \Phi_n K_n \log(K_n) \to 0$. It is asymptotically separable with $LMIT_{a,\gamma}$ for $\lambda_n K_n \Phi_n / \log(K_n) \to \infty$. The agnostic $LMIT_A$ achieves asymptotic separability with $\lambda_n / (\log(K_n)\sqrt{\log n}) \to \infty$.*

**Gaussian Model:** We next consider agnostic tests for Gaussian model of Eq. 3 with no propagation loss, i.e., $\alpha_{uv} = 1$.

**Theorem 7.** *The two hypotheses $H_0$ and $H_1$ for the Gaussian model are asymptotically inseparable if $\mu_n \Phi_n K_n \log(K_n) \to 0$, are separable with $LMIT_{a,\gamma}$ if $\mu_n K_n \Phi_n / \log(K_n) \to \infty$, and are separable with $LMIT_A$ if $\mu_n / (\log(K_n)\sqrt{\log n}) \to \infty$*

Our inseparability bound matches existing results on 2-D Lattice & Line Graphs by plugging in appropriate values for $\Phi$ for the cases considered in [2, 12]. The lower bound is obtained by specializing to a collection of "non-decreasing band" subgraphs. Yet $\text{LMIT}_{a,\gamma}$ and $\text{LMIT}_A$ is able to achieves the lower bound within a logarithmic factor. Furthermore, our analysis extends beyond Poisson & Gaussian models and applies to general graph structures and models. The main reason is that our LMIT analysis is fairly general and provides an observation-dependent bound through convex duality. We briefly describe it here. Consider functions $\ell_S(x)$ that are positive, separable

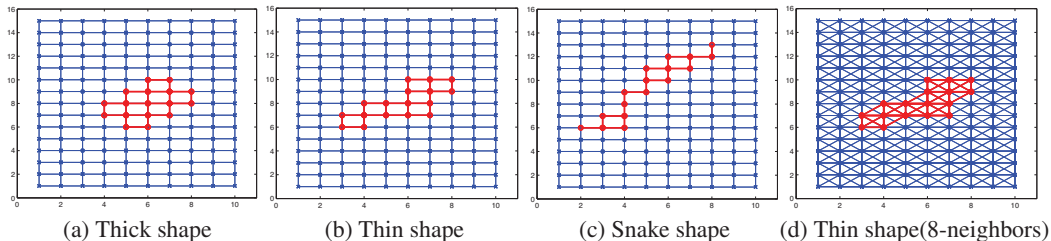

| (a) Thick shape | (b) Thin shape | (c) Snake shape | (d) Thin shape(8-neighbors) |

Figure 1: Various shapes of ground-truth anomalous clusters on a fixed 15×10 lattice. Anomalous cluster size is fixed at 17 nodes. (a) shows a thick cluster with a large internal conductance. (b) shows a relatively thinner shape. (c) shows a snake-like shape which has the smallest internal conductance. (d) shows the same shape of (b), with the background lattice more densely connected.

and bounded for simplicity. By establishing primal feasibility that the subgraph $S \in \mathcal{C}_{LMI}(a, \gamma)$ for a suitably chosen $\gamma$, we can obtain a lower bound for the alternative hypothesis $H_1$ and show that $E_{H_1} \left( \max_{M \in \mathcal{C}_{LMI}(a,\gamma)} \ell_M(x) \right) \geq E_{H_1} \left( \sum_{v \in S} \ell_S(x_v) \right)$. On the other hand for the null hypothesis we can show that, $E_{H_0} \left( \max_{M \in \mathcal{C}_{LMI}(a,\gamma)} \ell_M(x) \right) \leq E_{H_0} \left( \sum_{v \in B(a, \Theta(\sqrt{\gamma}))} \ell_S(x_v) \right)$. Here $E_{H_1}$ and $E_{H_0}$ denote expectations with respect to alternative and null hypothesis and $B(a, \Theta(\sqrt{\gamma}))$ is a ball-like thick shape centered at $a \in V$ with radius $\Theta(\sqrt{\gamma})$. Our result then follows by invoking standard concentration inequalities. We can extend our analysis to the non-separable case such as correlated models because of the linear objective form in Eq. 10.

## 5 Experiments

We present several experiments to highlight key properties of LMIT and to compare LMIT against other state-of-art parametric and non-parametric tests on synthetic and real-world data. We have shown that agnostic LMIT is near minimax optimal in terms of asymptotic separability. However, separability is an asymptotic notion and only characterizes the special case of zero false alarms (FA) and missed detections (MD), which is often impractical. It is unclear how LMIT behaves with finite size graphs when FAs and MDs are prevalent. In this context incorporating priors could indeed be important. Our goal is to highlight how shape prior (in terms of thick, thin, or arbitrary shapes) can be incorporated in LMIT using the parameter $\gamma$ to obtain better AUC performance in finite size graphs. Another goal is to demonstrate how LMIT behaves with denser graph structures.

From the practical perspective, our main step is to solve the following SDP problem:

$$\max_{M} : \sum_i y_i M_{ii} \qquad s.t. \ \ M \in \mathcal{C}_{LMI}(a, \gamma), \ \ tr(M) \leq K$$

We use standard SDP solvers which can scale up to $n \sim 1500$ nodes for sparse graphs like lattice and $n \sim 300$ nodes for dense graphs with $m = \Theta(n^2)$ edges.

To understand the impact of shape we consider the test LMIT$_{a,\gamma}$ for Gaussian model and manually vary $\gamma$. On a 15×10 lattice we fix the size (17 nodes) and the signal strength $\mu \sqrt{|S|} = 3$, and consider three different shapes (see Fig. 1) for the alternative hypothesis. For each shape we synthetically simulate 100 null and 100 alternative hypothesis and plot AUC performance of LMIT as a function of $\gamma$. We observe that the optimum value of AUC for thick shapes is achieved for large $\gamma$ and small $\gamma$ for thin shape confirming our intuition that $\gamma$ is a good surrogate for shape. In addition we notice that thick shapes have superior AUC performance relative to thin shapes, again confirming intuition of our analysis.

To understand the impact of dense graph structures we consider performance of LMIT with neighborhood size. On the lattice of the previous experiment we vary neighborhood by connecting each node to its 1-hop, 2-hop, and 3-hop neighbors to realize denser structures with each node having 4, 8 and 12 neighbors respectively. Note that all the different graphs have the same vertex set. This is convenient because we can hold the shape under the alternative fixed for the different graphs. As before we generate 100 alternative hypothesis using the thin set of the previous experiment with the same mean $\mu$ and 100 nulls. The AUC curves for the different graphs highlight the fact that higher density leads to degradation in performance as our intuition with complete graphs suggests. We also

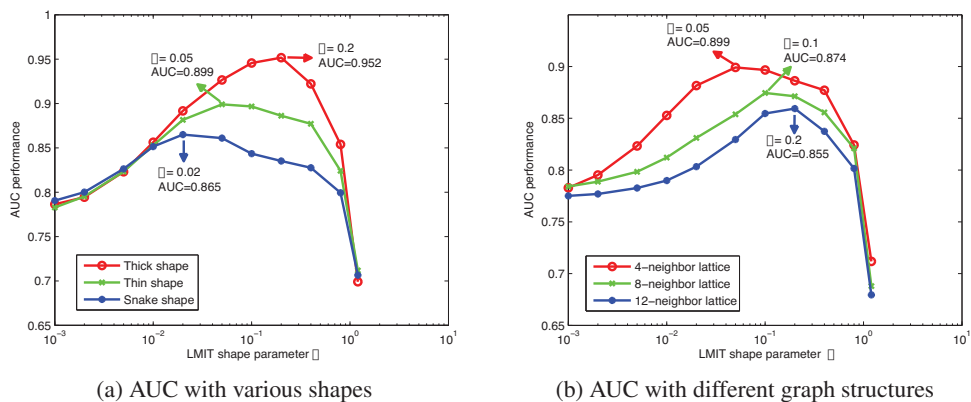

| (a) AUC with various shapes | (b) AUC with different graph structures |

Figure 2: (a) demonstrates AUC performances with fixed lattice structure, signal strength $\mu$ and size (17 nodes), but different shapes of ground-truth clusters, as shown in Fig.1. (b) demonstrates AUC performances with fixed signal strength $\mu$, size (17 nodes) and shape (Fig.1(b)), but different lattice structures.

see that as density increases a larger $\gamma$ achieves better performance confirming our intuition that as density increases the internal conductance of the shape increases.

In this part we compare LMIT against existing state-of-art approaches on a 300-node lattice, a 200-node random geometric graph (RGG), and a real-world county map graph (129 nodes) (see Fig.3,4). We incorporate shape priors by setting $\gamma$ (internal conductance) to correspond to thin sets. While this implies some prior knowledge, we note that this is not necessarily the optimal value for $\gamma$ and we are still agnostic to the actual ground truth shape (see Fig.3,4). For the lattice and RGG we use the elevated-mean Gaussian model. Following [1] we adopt an elevated-rate independent Poisson model for the county map graph. Here $N_i$ is the population of county, $i$. Under null the number of cases at county $i$, follows a Poisson distribution with rate $N_i\lambda_0$ and under the alternative a rate $N_i\lambda_1$ within some connected subgraph. We assume $\lambda_1 > \lambda_0$ and apply a weighted version of LMIT of Eq. 12, which arises on account of differences in population. We compare LMIT against several other tests, including simulated annealing (SA) [4], rectangle test (Rect), nearest-ball test (NB), and two naive tests: maximum test (MaxT) and average test (AvgT). SA is a non-parametric test and works by heuristically adding/removing nodes toward a better normalized GLRT objective while maintaining connectivity. Rect and NB are parametric methods with Rect scanning rectangles on lattice and NB scanning nearest-neighbor balls around different nodes for more general graphs (RGG and county-map graph). MaxT & AvgT are often used for comparison purposes. MaxT is based on thresholding the maximum observed value while AvgT is based on thresholding the average value.

We observe that uniformly MaxT and AvgT perform poorly. This makes sense; It is well known that MaxT works well only for alternative of small size while AvgT works well with relatively large sized alternatives [11]. Parametric methods (Rect/NB) performs poorly because the shape of the ground truth under the alternative cannot be well-approximated by Rectangular or Nearest Neighbor Balls. Performance of SA requires more explanation. One issue could be that SA does not explicitly incorporate shape and directly searches for the best GLRT solution. We have noticed that this has the tendency to amplify the objective value of null hypothesis because SA exhibits poor "regularization" over the shape. On the other hand LMIT provides some regularization for thin shape and does not admit arbitrary connected sets.

Table 1: AUC performance of various algorithms on a 300-node lattice, a 200-node RGG, and the county map graph. On all three graphs LMIT significantly outperforms the other tests consistently for all SNR levels.

| SNR | lattice ($\mu\sqrt{|S|}/\sigma$) | | | RGG ($\mu\sqrt{|S|}/\sigma$) | | | map ($\lambda_1/\lambda_0$) | | |
|---|---|---|---|---|---|---|---|---|---|
| | 1.5 | 2 | 3 | 1.5 | 2 | 3 | 1.1 | 1.3 | 1.5 |
| LMIT | **0.728** | **0.780** | **0.882** | **0.642** | **0.723** | **0.816** | **0.606** | **0.842** | **0.948** |
| SA | 0.672 | 0.741 | 0.827 | 0.627 | 0.677 | 0.756 | 0.556 | 0.744 | 0.854 |
| Rect(NB) | 0.581 | 0.637 | 0.748 | 0.584 | 0.632 | 0.701 | 0.514 | 0.686 | 0.791 |
| MaxT | 0.531 | 0.547 | 0.587 | 0.529 | 0.562 | 0.624 | 0.525 | 0.559 | 0.543 |
| AvgT | 0.565 | 0.614 | 0.705 | 0.545 | 0.623 | 0.690 | 0.536 | 0.706 | 0.747 |

## Footnotes

[1] A connected subgraph on a 2-D lattice of size $K$ has out-degree at least $\Omega(\sqrt{K})$ while set of subgraphs with out-degree $\Omega(\sqrt{K})$ includes disjoint union of $\Omega(\sqrt{K}/4)$ nodes. So statistical requirements with out-degree constraints can be no better than those for arbitrary $K$-sets.

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
