[Supplementary Material]

## Appendix: Proofs of Theorems

Figure 3: 300-node lattice and 200-node RGG with 17-node anomalous cluster.

Figure 4: County map, graph representation and ground truth anomalous cluster for Table 1.

**Proof of Theorem 3:**

*Proof.* For the first part we show $\forall a \in V, \gamma > 0, \mathcal{C}_{LMI}(a, \gamma) \subseteq \mathcal{C}$. Let $H = (S, F_S) \in \mathcal{C}_{LMI}(a, \gamma)$ be a connected subgraph. Assume on the contrary that $H$ is disconnected: $S = C \cup \bar{C}$, where $\bar{C} = S - C$. Let $|S| = k, |C| = k_1, |\bar{C}| = k_2$. W.l.o.g. assume $a = 1$, i.e. $M_{11} = 1$, and $C$ consists of nodes $\{1, 2, ..., k_1\}$.

Let $Q(M; \gamma) = L(A \circ M) - \gamma L(M)$. Consider the $k \times k$ sub-matrix $Q_S$ of $Q$ corresponding to $S$, since the rest part are all 0. Now we use the vector $g = [\mathbf{1}_{k_1}; -\mathbf{1}_{k_2}]$ to hit $Q_S$:

$$g'Q_S g = g'L_S(A_S \circ M_S)g - \gamma g'L_S(M_S)g \geq 0. \tag{13}$$

Note that $A_S$ has the form:

$$A_S = \begin{pmatrix} A_C & 0 \\ 0 & A_{\bar{C}} \end{pmatrix}, \tag{14}$$

where the off-diagonal block is zero because by assumption $C$ and $\bar{C}$ is disconnected. Then:

$$L_S(A_S \circ M_S) = Diag\left((A_S \circ M_S)\mathbf{1}_n\right) - (A_S \circ M_S) = \begin{pmatrix} \tilde{L}_C & 0 \\ 0 & \tilde{L}_{\bar{C}} \end{pmatrix}, \qquad (15)$$

where $\tilde{L}_C$ is the Laplacian matrix of $C$ weighted by $M_C$. Notice it still holds that $\tilde{L}_C \mathbf{1}_{k_1} = 0$. This means $g' L_S(A_S \circ M_S)g = 0$.

On the other hand, let $L_S(M_S)$ be:

$$L_S(M_S) = Diag\left(M_S \mathbf{1}_n\right) - M_S = \begin{pmatrix} L_1 & L_3 \\ L_3' & L_2 \end{pmatrix}. \qquad (16)$$

Using $g_1 = [\mathbf{1}_{k_1}; 0]$ and $g_2 = [0; \mathbf{1}_{k_2}]$ to hit $Q_S$ will yield: $\mathbf{1}_{k_1}' L_1 \mathbf{1}_{k_1} = 0$ and $\mathbf{1}_{k_2}' L_2 \mathbf{1}_{k_2} = 0$. Apparently $g' L_S(M_S)g \geq 0$ due to positive semi-definiteness of Laplacian matrix. If it's strictly positive, proof is done. Otherwise this means $\mathbf{1}_{k_1}' L_3 \mathbf{1}_{k_2} = 0$. Note that all entries of $L_3$ are either 0 or negative due to non-negativity of $M_S$. This means $L_3 = 0$, or equivalently $M_{ij} = 0$ for any $i \in C, j \in \bar{C}$. But this can not happen, because $M_{11} = 1$ and $M_{1j} \geq 1 + M_{jj} - 1 = M_{jj} > 0$ for any $j \in \bar{C}$. Contradiction! So $S$ is connected.

For the other direction we need to show that any connected subgraph $H = (S, F_S) \subseteq G = (V, E)$ has a corresponding matrix $H \rightleftharpoons M$, such that $M \in \mathcal{M}_a^*$ and $Q(M; \gamma) \succeq 0$ for some $a \in S$ and $\gamma > 0$.

Let $M$ be defined as:

$$M_{ij} = \begin{cases} 1 & i \in S, j \in S \\ 0 & \text{otherwise} \end{cases}$$

This $M$ can be viewed as the adjacency matrix corresponding to a complete graph on the node set $S$. So it naturally involves a star graph centered at $a$, and satisfies the linear constraints of $\mathcal{M}_a^*$.

Furthermore, the sub-block corresponding to $S$, $A_S \circ M_S$, is exactly the adjacency matrix of $H$. Since $H = (S, F_S)$ is connected, the second smallest eigenvalue of $L_S(A_S \circ M_s)$ is strictly positive. Notice that on the sub-block, $M_S = \mathbf{1}_k \mathbf{1}_k'$. Again by Finsler's Lemma, this means that there exists a $\gamma > 0$, such that the LMI holds on the sub-block:

$$L_S(A_S \circ M_S) - \gamma L(M_S) \succeq 0$$

$\square$

**Proof of Theorem 4:**

*Proof.* For simplicity we provide a proof sketch for rectangle bands on a 2D lattice $G$. We need to show that for a band $H = (S, F_S)$ belonging to $\mathcal{C}_{a,\Phi}$, there exists a binary matrix $M \rightleftharpoons H$ such that $L(A \circ M) - \gamma L(M) \succeq 0$, where $\gamma$ depends only on $\Phi$.

Construct the matrix $M$ as follows:

$$M_{ii} = \begin{cases} 1 & i \in S \\ 0 & otherwise \end{cases} , \quad M_{ij} = \begin{cases} 1 & (i,j) \in E_S \text{ or } i = a \text{ or} j = a \\ 0 & otherwise \end{cases}$$

Apparently $H \rightleftharpoons M$, and $M \in \mathcal{M}_a^*$. W.l.o.g. assume $a = 1$, and $S = \{1, 2, ..., k\}$. We only need to consider the first $k \times k$ sub-block of $Q(M; \gamma)$, denoted by $Q_S(M_S; \gamma) = L(A_S \circ M_S) - \gamma L(M_S)$. Notice $L(A_S \circ M_S)$ is exactly the unnormalized Laplacian matrix of $H = (S, F_S)$, and $L(M_S)$ is the Laplacian of the union graph of $H$ and $H_{star}$, where $H_{star}$ denote the star graph centered at node $a$.

Let $M_S = A_S \circ M_S + M_\triangle$. $M_\triangle$ is the adjacency matrix of a graph $H_\triangle$, where $H_\triangle$ is obtained from $H_{star}$ by removing those edges connected with the anchor. We rewrite the required inequality:

$$Q_S(M_S; \gamma) = L(A_S \circ M_S) - \gamma L(M_S) = (1 - \gamma)L(A_S \circ M_S) - \gamma L(M_\triangle) \succeq 0$$

Since $H_\triangle$ is obtained from $H_{star}$ by removing edges, we have $L(M_{star}) \succeq L(M_\triangle)$. We will show $\gamma = O(1/k) < 1/2$, which implies $\frac{\gamma}{1-\gamma} < 2\gamma$. Therefore it suffices to show:

$$L(A_S \circ M_S) - 2\gamma L(M_{star}) \succeq 0.$$

The rest part follows from Lemma 8, which characterizes the value of $\gamma$ for the above LMI to hold. Proof is done. □

**Lemma 8.** *Let $G = (V, E)$ denote a $k$-node rectangle band with width $a$ and length $b$ on the 2D lattice, i.e. $ab = k$. Let L be the graph Laplacian matrix corresponding to the rectangle lattice, and $L_{star}$ be the graph Laplacian of the star graph with the same node set, centered at the bottom-left node. Then the following inequality holds for $\gamma = \frac{\Phi^2}{4 \log(k\Phi)}$:*

$$L - \gamma L_{star} \succeq 0$$

*Proof.* Assume the anchor node is node 1. It is equivalent to show that for any $f \in \mathbb{R}^k$,

$$f'L_{star}f = \sum_{i \geq 2}(f_1 - f_i)^2 \leq \frac{1}{\gamma}f'Lf = \frac{1}{\gamma}\sum_{(i,j)\in E}(f_i - f_j)^2$$

We first investigate a simple case where $a = 1$, i.e. $G$ is a $k$-node line graph. In this scenario $\phi(G) = 2/k$. We use Cauchy-Schwartz inequality to bound each $(f_1 - f_i)^2$ using the edges on the path from node 1 to $i$:

$$(f_1 - f_i)^2 = \left(\sum_{j=1}^{i-1}(f_j - f_{j+1})\right)^2 \leq (i-1)\sum_{j=1}^{i-1}(f_j - f_{j+1})^2$$

Summing over all $(f_1 - f_i)^2$, we have:

$$\sum_{i=2}^{k}(f_1 - f_i)^2$$

$$\leq \sum_{i=2}^{k}\left[(i-1)\sum_{j=1}^{i-1}(f_j - f_{j+1})^2\right]$$

$$= \left(\sum_{i=1}^{k-1}i\right)(f_1 - f_2)^2 + \left(\sum_{i=2}^{k-1}i\right)(f_2 - f_3)^2 + ... + (k-1)(f_{k-1} - f_k)^2$$

$$\leq \frac{k^2}{2}\sum_{j=1}^{k-1}(f_j - f_{j+1})^2$$

Therefore the inequality for line graph holds.

Now w.l.o.g. assume $a \leq b$ and $a = 2^p$. We first show that to cover the $a^2/2$ nodes in the lower triangle, $\gamma = O(p2^p) = O(a^2 \log a)$ is enough. The strategy is similar: construct paths from anchor to each node, and apply Cauchy-Schwartz inequality to make use of edges on these paths. Two tricks need to be mentioned:
(1) Paths need to be constructed very carefully so that each edge of $G$ is not used too often;
(2) It is inevitable that some edges will be used much more frequently than others, for example, the edges coming out of anchor. A weighted Cauchy-Schwartz should therefore be applied to alleviate this effect.

Let each node be indexed by its coordinates, $(0,0)$ is the anchor node. To help understand the construction, we introduce several notations. A node $v = (x, y)$ is "critical" if $x + y = 2^q - 1$ for some integer $q$, as marked by red solid circles in Fig.5. Let $\mathbb{C}_q = \{v = (x,y)|x + y = 2^q - 1\}$ denote the collection of nodes on the $q$-th "boundary". Anchor node $v_0 = (0,0)$ is the only node in $\mathbb{C}_0$, and the outer most boundary is $\mathbb{C}_p$. Apparently $|\mathbb{C}_q| = 2^q$.

We build a complete balanced binary tree based on all critical nodes with tree edges $(v_i, v_{i+1})$, where $v_i \in \mathbb{C}_i$ denotes a critical node in $\mathbb{C}_i$. We note down several observations for paths from anchor to each $v_p \in \mathbb{C}_p$:
(1) There is a unique path starting from anchor $v_0 \in \mathbb{C}_0$ to each $v_p \in \mathbb{C}_p$, passing through critical nodes $v_i \in \mathbb{C}_i$, for $i = 0, 1, ..., p$.

Figure 5: Paths constructed to cover each node from anchor.

(2) Such a path, denoted by $v_0 \to v_1 \to ... \to v_p$ where $v_i \in \mathbb{C}_i$, is composed of $p$ tree edges, $(v_i, v_{i+1})$ for $i = 0, 1, ..., p-1$, with $|(v_i, v_{i+1})| = 2^i$.

(3) For any two such paths, after they split at some node, they will never share any graph edges.

Now consider a path from $v_0$ to some $v_p \in \mathbb{C}_p$, $v_0 \to v_1 \to ... \to v_p$. We use weighted Cauchy-Schwartz inequality to bound this path with graph edges:

$$
\left(f_{v_0} - f_{v_p}\right)^2
$$

$$
= \left(\sum_{i=0}^{p-1}(f_{v_i} - f_{v_{i+1}})\right)^2
$$

$$
= \left((f_{v_0} - f_{v_1}) + \sum_{(i,j)\in(v_1,v_2)}(f_i - f_j) + ... + \sum_{(i,j)\in(v_{p-1},v_p)}(f_i - f_j)\right)^2
$$

$$
\leq (1 \times 2^{p-1} + 2 \times 2^{p-2} + ... + 2^{p-1} \times 1)
$$
$$
\cdot \left(\frac{(f_{v_0} - f_{v_1})^2}{2^{p-1}} + \frac{\sum_{(i,j)\in(v_1,v_2)}(f_i - f_j)^2}{2^{p-2}} + ... + \frac{\sum_{(i,j)\in(v_{p-1},v_p)}(f_i - f_j)^2}{1}\right)
$$

$$
= p\left((f_{v_0} - f_{v_1})^2 + 2\sum_{(i,j)\in(v_1,v_2)}(f_i - f_j)^2 + ... + 2^{p-1}\sum_{(i,j)\in(v_{p-1},v_p)}(f_i - f_j)^2\right)
$$

The intuitive idea is that the graph edges composing tree edges closer to the anchor, i.e. $(i, j) \in (v_l, v_{l+1})$ for small $l$ where $v_l \in \mathbb{C}_l$, will be passed through many more times than those composing tree edges far away from the anchor. So when applying weighted Cauchy-Schwartz inequality, a larger denominator is imposed on $(f_i - f_j)^2$ for those $(i, j) \in (v_l, v_{l+1})$ for small $l$. For example, for the most frequently used edge $(v_0, v_1)$, a penalty of $2^{p-1}$ is imposed on these edges (2 such edges, ((0,0),(0,1)) and ((0,0),(1,0))), while for those graph edges composing $(v_{p-1}, v_p)$, only a constant is put in the denominator.

Next we need to figure out the frequency that each graph edge is used for covering all the nodes. By induction it is not hard to observe that the graph edges on the tree edge $(i, j) \in (v_l, v_{l+1})$ will

be passed by at most $2^{2p-1-l}$ paths. Take the graph of Fig.5 as an example. Each path is of the form $v_0 \to ... \to v_4$, $v_i \in \mathbb{C}_i$. The edges on $(v_3, v_4)$ are used at most 8 times, eg. $((7,0),(8,0))$. We have $8 < 16 = 2^{2p-1-3}$. The edges on $(v_2, v_3)$ are used at most $8 \times 2 + 4 = 20$ times, eg. $((3,0),(4,0))$. $20 < 32 = 2^{2p-1-2}$. The edges on $(v_1, v_2)$ are used at most $20 \times 2 + 2 = 42$ times, eg. $((1,0),(2,0))$. $42 < 64 = 2^{2p-1-1}$. The top-most edges, $((0,0),(1,0))$ and $((0,0),(0,1))$, are used $42 \times 2 + 1 = 85$ times. $85 < 128 = 2^{2p-1-0}$.

So summing over all paths from anchor to all nodes within the lower triangle $T$:

$$\sum_{v \in T} (f_{v_0} - f_v)^2$$

$$\leq \quad p \sum_{v_0 \to ... \to v_p \in \mathbb{C}_p} \left( 2^{2p-1}(f_{v_0} - f_{v_1})^2 + ... + 2^{2p-1)} \sum_{(i,j) \in (v_{p-1}, v_p)} (f_i - f_j)^2 \right)$$

$$\leq \quad p 2^{2p-1} \sum_{(i,j) \in E} (f_i - f_j)^2$$

Note that $p 2^{2p-1} = a^2 \log a / 2$. So:

$$\gamma = \frac{2}{a^2 \log a}$$

is enough to cover all nodes in the lower triangle of an $a \times b$ rectangle lattice as in Fig.(5).

To cover the rest nodes, i.e. blue nodes in Fig.5, we build paths that horizontally extend from the outer-most boundary nodes $v_p \in \mathbb{C}_p$. Let $v_{p'}$ denote the rightmost node extending horizontally from $v_p \in \mathbb{C}_p$. Similarly we use weighted Cauchy-Schwartz inequality to bound the path: $v_0 \to ... \to v_p \to v_{p'}$:

$$\left( f_{v_0} - f_{v_{p'}} \right)^2$$

$$= \quad \left( (f_{v_0} - f_{v_1}) + ... + \sum_{(i,j) \in (v_{p-1}, v_p)} (f_i - f_j) + \sum_{(i,j) \in (v_p, v_{p'})} (f_i - f_j) \right)^2$$

$$\leq \quad \left( 1 \times 2^{p-1} + 2 \times 2^{p-2} + ... + 2^{p-1} \times 1 + b \times 1 \right)$$
$$\cdot \left( \frac{(f_{v_0} - f_{v_1})^2}{2^{p-1}} + ... + \frac{\sum_{(i,j) \in (v_{p-1}, v_p)} (f_i - f_j)^2}{1} + \frac{\sum_{(i,j) \in (v_p, v_{p'})} (f_i - f_j)^2}{1} \right)$$

$$= \quad (p 2^{p-1} + b) \left( \sum_{l=0}^{p-1} \frac{\sum_{(i,j) \in (v_l, v_{l+1})} (f_i - f_j)^2}{2^{p-1-l}} + \sum_{(i,j) \in (v_p, v_{p'})} (f_i - f_j)^2 \right)$$

It is easy to observe that to cover these extended nodes, the graph edges $(i,j) \in (v_l, v_{l+1})$ are passed through $b 2^{p-1-l}$ times for $l = 0, 1, ..., p-1$, and $b$ times for those extended edges $(i,j) \in (v_p, v_{p'})$. Now totally we have:

$$\sum_v (f_{v_0} - f_v)^2 \leq \left( p 2^{2p-1} + b(p 2^{p-1} + b) \right) \sum_{(i,j) \in E} (f_i - f_j)^2$$

Plugging in $2^p = a$, $a \leq b$ and $ab = k$, we have:

$$\sum_v (f_{v_0} - f_v)^2 \quad \leq \quad \left( ab \log a + b^2 \right) \sum_{(i,j) \in E} (f_i - f_j)^2$$

$$\leq \quad \max \left( 2k \log \frac{k}{b}, 2b^2 \right) \sum_{(i,j) \in E} (f_i - f_j)^2$$

Note that $\Phi = \frac{a}{k/2} = \frac{2}{b}$. Replace $b$ with $\Phi$, the proof is done.

We list two extreme examples for demonstration. For the thinnest line graph where $a = 1, b = k$ and $\Phi = 2/k$, $\gamma = \frac{1}{2k^2} = \Phi^2/8$ is sufficient to have: $L - \gamma L_{star} \succeq 0$. For the other extreme case where the graph is a square lattice with $a = b = \sqrt{k}$, $\Phi = 2/\sqrt{k}$, $\gamma = \frac{1}{k \log k\Phi} = \Phi^2/4 \log(k\Phi)$ is required for the LMI to hold. Note that $\Phi$ is between $O(1/\sqrt{k})$ and $\Omega(k)$. So at least the smaller $\gamma = \Theta(\Phi^2/\log k)$ can make the LMI hold. Proof is done.

$\square$

For future use we present the explicit form of the dual problem to a primal problem that has constraints $M \in \mathcal{C}_{LMI}(a, \gamma)$. Interestingly, the dual problem corresponds to finding an embedding of all nodes in a 1D Euclidean space, such that certain constraints at each node and edge of the graph hold.

**Lemma 9.** *Given $G = (V, E)$ with adjacency matrix $A$, let $y_i$ denote the variable associated with node $i \in V$. Assume w.l.o.g. the anchor is node 1. Consider the following SDP problem, where the constraints are exactly those of $M \in \mathcal{C}_{LMI}(1, \gamma)$:*

$$\text{max}: \quad \sum_i y_i M_{ii} \tag{17}$$

$$\text{s.t.} \quad Q(M; \gamma) = L(A \circ M) - \gamma L(M) \succeq 0$$
$$M_{ij} \geq 0, \quad \forall 2 \leq i < j$$
$$1 - M_{ii} \geq 0, \quad \forall 2 \leq i$$
$$M_{ii} - M_{ij} \geq 0, \quad \forall 2 \leq i < j$$
$$M_{jj} - M_{ij} \geq 0, \quad \forall 2 \leq i < j$$

*Then the corresponding dual problem has the following form:*

$$\text{min}: \quad y_1 + \sum_{i \geq 2} \rho_i \tag{18}$$

$$\text{s.t.} \quad y_i + (1 - \gamma) z_i^2 + \sum_{2 \leq j \neq i, (i,j) \in E} \alpha_{ij} + \alpha_i = \rho_i, \quad \forall i \geq 2, (1, i) \in E$$

$$y_i - \gamma z_i^2 + \sum_{2 \leq j \neq i, (i,j) \in E} \alpha_{ij} + \alpha_i = \rho_i, \quad \forall i \geq 2, (1, i) \notin E$$

$$(1 - \gamma)(z_i - z_j)^2 \leq \alpha_{ij} + \alpha_{ji}, \quad \forall 2 \leq i < j, (i, j) \in E$$

$$\rho_i \geq 0, \alpha_{ij} \geq 0, \alpha_i \geq 0, z_i \geq 0$$

*where $z_i$, a scalar dual variable, is the embedding coordinate of node $i \geq 2$; the rest dual variables include $\alpha_i, \rho_i, \forall i \geq 2$ and $\alpha_{ij}, \forall (i, j) \in E$.*

*Proof.* The explicit Lagrangian of Eq.(17) is:

$$L = y_1 + \sum_{i \geq 2} M_{ii} y_i + \langle Q, G \rangle + \sum \sum_{2 \leq i < j} \mu_{ij} M_{ij} + \sum_{i \geq 2} \rho_i (1 - M_{ii}) \tag{19}$$

$$+ \sum \sum_{2 \leq i < j} \alpha_{ij} (M_{ii} - M_{ij}) + \sum \sum_{2 \leq j < i} \alpha_{ij} (M_{ii} - M_{ji})$$

where $G \succeq 0, \mu_{ij} \geq 0, \rho_i \geq 0, \alpha_{ij} \geq 0$ are lagrange multipliers. Notice the symmetric matrix $Q$ can be decomposed into the following form:

$$Q(M; \gamma) = L(A \circ M) - \gamma L(M) = \sum \sum_{i < j} \left( \mathbf{1}_{(i,j)} - \gamma \right) M_{ij} (e_{ii} + e_{jj} - e_{ij} - e_{ji})$$

where $\mathbf{1}_{(i,j)}$ is the indicator of $(i, j) \in E$, $e_{ij}$ denotes the matrix with value 1 at $(i, j)$ and 0 elsewhere. Plugging in $M_{1i} = M_{ii}$, we have:

$$\langle Q, G \rangle = \sum_{i \geq 2} \left( \mathbf{1}_{(1,i)} - \gamma \right) M_{ii} (G_{11} + G_{ii} - 2G_{1i})$$

$$+ \sum \sum_{2 \leq i < j} \left( \mathbf{1}_{(i,j)} - \gamma \right) M_{ij} (G_{ii} + G_{jj} - 2G_{ij})$$

Taking derivatives w.r.t. $M_{ii}$ and $M_{ij}$ respectively, the dual problem is:

$$\min : \quad y_1 + \sum_{i \geq 2} \rho_i \tag{20}$$

$$s.t. \quad y_i + \left(1_{(1,i)} - \gamma\right) G_{(1i)} + \sum_{2 \leq j \neq i} \alpha_{ij} = \rho_i, \ \forall i \geq 2$$

$$\left(1_{(i,j)} - \gamma\right) G_{(ij)} + \mu_{ij} - \alpha_{ij} - \alpha_{ji} = 0, \ \forall 2 \leq i < j$$

$$G \succeq 0, \mu_{ij} \geq 0, \rho_i \geq 0, \alpha_{ij} \geq 0$$

where $G_{(ij)} = G_{ii} + G_{ii} - 2G_{ij}$.

Since $G$ is symmetric and PSD, we have $G = VV'$ such that $G_{ij} = v_i'v_j$. $v_i \in \mathbb{R}^n$ can be viewed as the embedding of node $i$ in the $n$-dimensional Euclidean space. $G_{(ij)} = ||v_i - v_j||^2$ is simply the squared distance between the embeddings of node $i$ and $j$. We write constraints separately based on indicators:

$$\min : \quad y_1 + \sum_{i \geq 2} \rho_i \tag{21}$$

$$s.t. \quad y_i + (1 - \gamma) ||v_i - v_1||^2 + \sum_{2 \leq j \neq i} \alpha_{ij} = \rho_i, \ \forall i \geq 2, (1, i) \in E$$

$$y_i - \gamma ||v_i - v_1||^2 + \sum_{2 \leq j \neq i} \alpha_{ij} = \rho_i, \ \forall i \geq 2, (1, i) \notin E$$

$$(1 - \gamma) ||v_i - v_j||^2 + \mu_{ij} - \alpha_{ij} - \alpha_{ji} = 0, \ \forall 2 \leq i < j, (i, j) \in E$$

$$-\gamma ||v_i - v_j||^2 + \mu_{ij} - \alpha_{ij} - \alpha_{ji} = 0, \ \forall 2 \leq i < j, (i, j) \notin E$$

$$\mu_{ij} \geq 0, \rho_i \geq 0, \alpha_{ij} \geq 0$$

We further simplify this dual formulation. Notice that for constraints of $(i, j) \notin E$, $\mu_{ij} \geq 0$ is an independent and completely free variable that can always make such a constraint hold. So we can drop these redundant constraints. For edge constraints of $(i, j) \in E$, we replace $\mu_{ij}$ with inequalities. For node constraints of node $i$, we split out those $\alpha_{ij}$ with $(i, j) \notin E$ which are independent and combine them into a new variable $\alpha_i \geq 0$. Also note that the embedding of anchor, $v_1$, is completely free variable, which we can fix w.l.o.g. at 0. The dual problem is simplified as follows:

$$\min : \quad y_1 + \sum_{i \geq 2} \rho_i \tag{22}$$

$$s.t. \quad y_i + (1 - \gamma) ||v_i||^2 + \sum_{2 \leq j \neq i, (i,j) \in E} \alpha_{ij} + \alpha_i = \rho_i, \ \forall i \geq 2, (1, i) \in E$$

$$y_i - \gamma ||v_i||^2 + \sum_{2 \leq j \neq i, (i,j) \in E} \alpha_{ij} + \alpha_i = \rho_i, \ \forall i \geq 2, (1, i) \notin E$$

$$(1 - \gamma) ||v_i - v_j||^2 \leq \alpha_{ij} + \alpha_{ji}, \ \forall 2 \leq i < j, (i, j) \in E$$

$$\rho_i \geq 0, \alpha_{ij} \geq 0, \alpha_i \geq 0$$

Note the constraints have been divided into 3 categories: node constraints of those nodes directly linking to the anchor node, node constraints of the rest nodes, and edge constraints of edges among all nodes except the anchor.

The key observation is that each embedding vector $v_i$ only appears in node constraints with its length $||v_i||$, while only distances between embeddings exist in edge constraints, which are all inequalities. We perform several operations on $v_i$ while maintaining dual feasibility. The first step is to fold all $v_i$ into a fixed quadrant so that $||v_i||$ remains unchanged while $||v_i - v_j||$ either remains unchanged or is decreased. This can be done by first fixing a Euclidean coordinate system, with $n$ hyperplanes intersecting at 0 and pairwise perpendicular. Then for each such hyperplane that partitions the whole space into two half-spaces, we fold all $v_i$ in the "left" half-space to the "right" half-space axis-symmetrically. It is obvious that this folding operation maintains $||v_i||$ for all $i$ and $||v_i - v_j||$

for those $i, j$ in the same half-space. The rest $i, j$, $||v_i - v_j||$ are only decreased due to Pythagoras theorem. After folding for all these hyperplanes, all $v_i$ now locate in the same quadrant such that $v'_i v_j \geq 0$, $\forall i, j$, i.e. angles between $v_i$ and $v_j$ are smaller than $\pi/2$. Yet all node and edge constraints are still satisfied.

The second step is mapping all $v_i$ onto one single direction:

$$v_i \in \mathbb{R}^n \mapsto z_i \in \mathbb{R}^+ : \quad z_i = ||v_i||$$

By definition all node constraints are satisfied. Again by Pythagoras and the $\pi/2$ condition, $||v_i - v_j||$ is decreased so that edge constraints are satisfied. Therefore the dual problem Eq.(22) can be reduced to the equivalent Eq.(18). Proof is done.

$\square$

To prove the main theorem, we need the following lemma.

**Lemma 10.** *On a graph with maximum degree $D$, consider the following max-trace problem:*

$$\begin{aligned}
\max : \quad & tr(M) && (23) \\
s.t. \quad & L(A \circ M) - \gamma L(M) \succeq 0 \\
& M_{ij} = M_{ji}, \ M_{11} = 1, \ M_{1i} = M_{ii} \\
& 0 \leq M_{ij} \leq M_{ii}, M_{jj} \leq 1
\end{aligned}$$

*Let $M^* = M^*(\gamma)$ be the optimal solution to this problem. Then $M^*$ has the following properties:*

1. *$tr(M^*) \leq D/\gamma$, where $D$ is the max degree of the graph.*

2. *The node set $V_0 = \{i : M^*_{ii} = 1\}$, including the anchor, form a connected sub-graph.*

3. *The 1-hop outer layer, $V_1 = \{i : (i, j) \in E, j \in V_0, i \notin V_0\}$, satisfy: $0 \leq M^*_{ii} < 1$.*

4. *The rest nodes are: $M^*_{ii} = 0$.*

**Remark:**
This lemma is just saying that the solution $M^*$ to the max-trace problem has a nested structure centered at the anchor. The interior of the support of $diag(M^*)$ have value $M^*_{ii} = 1$, the boundary $0 \leq M^*_{ii} < 1$, and the rest nodes have $M^*_{ii} = 0$. We conjecture that $M^*$ always has a "fattest" shape. At least by Theorem 4 $M^*$ contains a square of size $\Theta(k)$ if $\gamma = \Theta(\frac{1}{k \log k})$. Fig.6 shows two solutions of the max-trace problem with different values of $\gamma$. Intuitively, smaller $\gamma$ allows the search to extend farther away than larger $\gamma$.

(a) $\gamma = 0.3$          (b) $\gamma = 0.08$

Figure 6: Optimal solution $M^*$ of max-trace problem, with large / small values of $\gamma$. Values of $M^*_{ii}$ are illustrated through grey-scale. Rped node is the anchor.

*Proof.* This is the problem of Eq.(17) with all $y_i = 1$. According to Eq.(18), the corresponding dual problem is:

$$\min : \quad 1 + \sum_{i \geq 2} \rho_i$$

$$s.t. \quad 1 + (1 - \gamma) z_i^2 + \sum_{2 \leq j \neq i, (i,j) \in E} \alpha_{ij} + \alpha_i = \rho_i, \ \forall i \geq 2, (1,i) \in E$$

$$1 - \gamma z_i^2 + \sum_{2 \leq j \neq i, (i,j) \in E} \alpha_{ij} + \alpha_i = \rho_i, \ \forall i \geq 2, (1,i) \notin E$$

$$(1 - \gamma) (z_i - z_j)^2 \leq \alpha_{ij} + \alpha_{ji}, \ \forall 2 \leq i < j, (i,j) \in E$$

$$\rho_i \geq 0, \alpha_{ij} \geq 0, \alpha_i \geq 0, z_i \geq 0$$

We show (1) by constructing a simple dual solution to yield an upper bound on the max-trace problem. Let $z_i = z, \forall i \geq 2$, so that all edge constraints automatically hold. Let $z^2 = 1/\gamma$, $\alpha_{ij} = \alpha_i = 0$, so that $\rho_i = 0, \forall i \geq 2, (1,i) \notin E$. The cost of this dual feasible solution, thus an upper bound on $tr(M^*)$, is:

$$tr(M^*) \leq \sum_{i:(1,i) \in E} \left( 1 + (1 - \gamma)z^2 \right) \leq D/\gamma$$

The intuition is that $z_i$ increases as $i$ goes farther away from the anchor, until $\gamma z_i^2 \geq y_i = 1$ for all nodes $i$ outside some closed layer $B$ which contains the anchor. This layer corresponding to the above trivial solution is simply the set of 1-hop neighbors of anchor: $B = \{i : (1,i) \in E\}$. But this dual feasible solution increases $z_i$ too fast (in one step), thus pays too much price at $\rho_i$ for these direct neighbors.

Let $V_0$ be the set of nodes with $M_{ii}^* = 1$ and connected to the anchor node 1. Let $V_1$ be the 1-hop outer layer of $V_0$, and $V_2$ the 1-hop outer layer of $V_1$. Since strong duality holds, by complementary slackness, the optimal dual variables have: $\rho_i = 0, \forall i \in V_1$. We create slackness for all edges between $V_1$ and $V_2$, which correspond to the original dual variables $\mu_{ij}$ back in Eq.(21). Again by complementary slackness, if $\mu_{ij} > 0$, then the primal $M_{ij} = 0$. We have disconnected nodes in $V_0$ from outside $V_1$. By Theorem 3 the support of $diag(M)$ is connected. So $M_{ii} = 0$ for those nodes outside $V_1$.

To create this slackness for edges between $V_1$ and $V_2$, consider a modified primal objective:

$$\max : \quad \sum_{i \in V_0 \cup V_1} M_{ii} + (1 - \epsilon) \sum_{i \notin V_0 \cup V_1} M_{ii} \tag{24}$$

$$s.t. \quad M \in \mathcal{C}_{LMI}(1, \gamma)$$

The optimal dual solution to the max-trace problem is also feasible for this modified problem, which gives the same dual cost. Now outside $V_1$:

$$(1 - \gamma) (z_i - z_j)^2 \leq \alpha_{ji} + \alpha_{ij}, \quad \forall j \in V_1, i \in V_2 \ (j,i) \in E$$

$$1 - \epsilon - \gamma z_i^2 + \sum_{2 \leq j \neq i, (i,j) \in E} \alpha_{ij} + \alpha_i = \rho_i, \ \forall i \in V_2$$

Leaving other dual variables unchanged, we can distribute $\epsilon$ uniformly on those $\alpha_{ij}$, $i \in V_2, j \in V_1$ to create the slackness we want on edges $(j,i)$. Proof is done.

$\square$

The proofs of main theorems for Poisson and Gaussian models follow similar lines. Here we only elaborate on the Gaussian case.

**Proof for Gaussian model:**

*Proof.* The proof consists of 2 parts:

- Inseparability: This part generalizes the results of [12] in terms of the internal conductance parameter $\Phi$ rather than the length and width used in [12]. This is shown in Lemma 11.

- Separability: This part itself can be divided into two steps.

  1. We first show under $H_0$ the optimal value of the test is upper bounded by using a modified version of $M^*$, the optimal solution to the max-trace problem. This is shown in Lemma 12.
  2. We then show that under $H_1$, the feasible solution $M^*$ to the max-trace problem covers a large portion of the ground-truth cluster for our problem.

By Lemma 12 we have:

$$c^*|_{H_0} \le N(0, tr(M^*)) + O\left(\sqrt{\frac{\log k}{\gamma}}\right)$$

For the $H_1$ case, for simplicity we consider a band $B$ of size $k$, with width $a$ and length $b$, $ab = k$. The corresponding conductance is $\Phi = \Theta(1/b)$. Such a band must be contained in a square of size $b \times b$, i.e. $\Theta(1/\Phi^2)$. On the other hand, for this band we choose $\gamma = \Theta(\Phi^2/\log k)$. The $M^*$ of the max-trace problem with this $\gamma$ at least contains a square of size $\Theta(1/\Phi^2)$. Therefore by appropriately positioning the anchor, $M^*$ overlaps $B$ at least on $\Theta(k)$ nodes. This means if we simply adopt $M^*$ as a primal feasible solution, we have:

$$c^*|_{H_1} \ge N(0, tr(M^*)) + \Theta(k)\mu$$

Note that $tr(M^*) = O(1/\gamma)$. To asymptotically separate $H_0$ and $H_1$, it suffices that:

$$tr(M^*) + O(\sqrt{1/\gamma}) + O(\sqrt{\log k/\gamma}) \le tr(M^*) - O(\sqrt{1/\gamma}) + \Theta(k)\mu,$$

where the terms $O(\sqrt{1/\gamma})$ on both sides correspond to the standard deviation term. Plugging in $\gamma = \Phi^2/log(k)$, we have:

$$\mu = \Omega\left(\frac{\log k}{k\Phi}\right)$$

When the anchor is unknown, applying the test for different anchors induces an additional $\sqrt{\log n}$ term due to union bound. When the shape is unknown, the test sets $\gamma$ according to the smallest conductance, i.e. $\gamma = \Theta(1/k^2)$, to search for the thinnest shape with size $k$. In this case, the requirement on $\mu$, when agnostic to anchor and shape, is:

$$\mu = \Omega\left(\log k \sqrt{\log n}\right)$$

Proof is done. □

**Lemma 11.** *The two hypothesis $H_0$ and $H_1$ are asymptotically inseparable if:*

$$\mu_n K_n \Phi_n \log(K_n) \to 0$$

*Proof.* The collection of anomalous subgraphs with size $K_n$ and internal conductance $\Phi_n$ contains the bands of width $h_n$ and length $l_n$ defined in Theorem 3 of [12]. So the inseparability result there also holds for our case. Roughly we have:

$$l_n h_n = K_n, \quad \frac{h_n}{K_n} = \Phi_n$$

By Theorem 3 in [12], $H_0$ and $H_1$ are asymptotically inseparable if: (ignoring the $\log\log()$ term)

$$\mu_n \sqrt{K_n} \left(\frac{l_n}{h_n}\right)^{-1/2} \log(l_n) \to 0$$

Substitute $l_n$ and $h_n$ using $K_n$ and $\Phi_n$, and note that $1/\Phi \ge \sqrt{K_n}$. We get:

$$\mu_n K_n \Phi_n \log(K_n) \to 0.$$

□

**Lemma 12.** *Assume $x_i$ follows standard normal distribution for all nodes $i$. The optimal cost of problem Eq.(17) with signal $x_i$ for node $i$ is upper bounded by:*

$$c^*|_{H_0} \leq \sum_i x_i M_{ii}^* + \Theta\left(\sqrt{\log\left(\frac{1}{\gamma}\right)/\gamma}\right)$$

*where $M^*$ is the optimal solution to the max-trace problem with parameter $\gamma$.*

*Proof.* Let $y_i = 1 + x_i/N$, where $N$ is a normalization constant to be decided. We show that for appropriately chosen $N$, the modified problem Eq.(17) with signal $y_i$ has the optimal cost with some upper bound. We then recover the original problem by first subtracting $tr(M^*)$, following by multiplying $N$.

Write $y_i = (1 + x_{max}/N) - (x_{max} - x_i)/N = (1 + x_{max}/N) - \eta_i$, where $x_{max} = \max_{i \in H^*(M^*)} |x_i|$, $\eta_i = (x_{max} - x_i)/N$. Note that $x_{max}$ scales as $\Theta(\sqrt{|H^*|})$ for i.i.d. standard normal random variables, where $H^*(M^*)$ is the resulting fat shape corresponding to the max-trace problem. Note that $0 \leq \eta_i \leq 2x_{max}/N$ for $i \in H^*$. Consider the dual solution of the max-trace problem. We know that for nodes $i \in V_0$ the dual variables $\rho_i > 0$. Let $\delta = \min_{i \in V_0} \rho_i > 0$, which is a constant depending only on $\gamma$. Consider the following problem:

$$\text{max}: \quad (1 + x_{max}/N)tr(M) \tag{25}$$
$$s.t. \quad M \in \mathcal{C}_{LMI}(1, \gamma)$$

Since $x_{max}$ is just a constant, the optimal dual solution to this problem is just the $(1 + x_{max}/N)$-stretched version of that of the max-trace problem. So $\min_{i \in V_0} \rho_i' = (1 + x_{max}/N)\delta > \delta$. Now choose $N$ sufficiently large such that

$$\eta_i \leq 2x_{max}/N \leq \delta < \min_{i \in V_0} \rho_i'$$

We modify this dual solution of Eq.(25) to build a dual feasible solution for:

$$\text{max}: \quad \sum_i y_i M_{ii} \tag{26}$$
$$s.t. \quad M \in \mathcal{C}_{LMI}(1, \gamma)$$

Let $\tilde{c}$ denote the optimal cost. By Lemma 9 the corresponding dual problem is:

$$\text{min}: \quad y_1 + \sum_{i \geq 2} \rho_i \tag{27}$$
$$s.t. \quad 1 + \frac{x_{max}}{N} - \eta_i + (1 - \gamma) z_i^2 + \sum_{2 \leq j \neq i, (i,j) \in E} \alpha_{ij} + \alpha_i = \rho_i, \ \forall i \geq 2, (1, i) \in E$$
$$1 + \frac{x_{max}}{N} - \eta_i - \gamma z_i^2 + \sum_{2 \leq j \neq i, (i,j) \in E} \alpha_{ij} + \alpha_i = \rho_i, \ \forall i \geq 2, (1, i) \notin E$$
$$(1 - \gamma)(z_i - z_j)^2 \leq \alpha_{ij} + \alpha_{ji}, \ \forall 2 \leq i < j, (i, j) \in E$$
$$\rho_i \geq 0, \alpha_{ij} \geq 0, \alpha_i \geq 0, z_i \geq 0$$

The only differences between Eq.(27) and the dual problem of Eq.(25) are those $-\eta_i$ at node constraints. Based on the dual optimal solution of Eq.(25), we modify dual variables to build a dual feasible solution for Eq.(27). Two cases need to be considered.

- For nodes $i \in V_0$, simply let $\rho_i' = \rho_i - \eta_i$. Note that we still have dual feasibility: $\rho_i' \geq 0$ by construction of $N$.

- For nodes $i \in H^* - V_0$ where $\rho_i' = \rho_i = 0$, we increase the free variables, $\alpha_i' = \alpha_i + \eta_i$, to absorb the difference, while keeping $\rho_i' = 0$ unchanged.

- For nodes outside $H^*$, since we know the size $k$, for ease of proof we simply zero out all $y_i$.

In this way we have built a dual feasible solution of Eq.(27). The corresponding dual cost, thus an upper bound on the primal optimum of Eq.(26) by weak duality, is:

$$
\begin{aligned}
\tilde{c} \ &\leq \ (1 + \frac{x_{max}}{N})tr(M^*) - \sum_{i \in V_0} \eta_i \\
&= \ tr(M^*) + \frac{x_{max}}{N}tr(M^*) - \frac{x_{max}}{N}|V_0| + \sum_{i \in V_0} \frac{x_i}{N} \\
&= \ tr(M^*) + \frac{x_{max}}{N}\beta|V_1| + \sum_{i \in V_0} \frac{x_i}{N} \\
&\leq \ tr(M^*) + \sum_i \frac{x_i M_{ii}^*}{N} + \frac{x_{max}}{N}\beta|V_1|
\end{aligned}
$$

where $\beta|V_1| = tr(M^*) - |V_0|$ is the fractional boundary part of $M^*$. This part can be 0 for some values of $\gamma$, or can be maximally $|V_1|$. Note that $H^*$ is a fat shape in lattice, so the boundary is: $|V_1| = \Theta\left(\sqrt{tr(M^*)}\right) = O(1/\sqrt{\gamma})$.

Since $y_i = 1 + x_i/N$, we restore the solution by subtracting $tr(M^*)$ and then multiplying $N$, which gives:

$$
c^*|_{H_0} \leq \sum_i x_i M_{ii}^* + x_{max}O(\sqrt{1/\gamma})
$$

$\square$

**Lemma 13.** $G = (V, E)$ *is a connected subgraph on an infinitely large 2D lattice. $G$ also satisfies:*

1. *$|V| = \Omega(k)$;*

2. *the conductance of $G$ is $\Theta(1/\sqrt{k})$:*

*Then $G$ must contain a triangle of size $\Theta(k)$.*

*Proof.* We provide an intuitive sketch. Consider all horizontal cuts on $G$. The most "balanced" horizontal cut $C_h$, where both parts are of size $\Theta(k)$, must have length $\Omega(\sqrt{k})$, otherwise (2) will be violated. Consider all vertical cuts within the range of the balanced horizontal cut range. Similar arguments follow that the most balanced vertical cut $C_v$ has size $\Omega(\sqrt{k})$.

Consider vertical cuts that start from $C_v$ and move aside stepwise along $C_h$. Assume at some step the vertical cut passes through $a$ edges, the smaller part has $b$ nodes, and the conductance here is tight: $\frac{a}{b} = \Phi = \Omega(1/\sqrt{k})$. For the next vertical cut, assume the cut decreases by $\delta$ edges. The conductance at the new vertical cut is: $\frac{a-\delta}{b-a} \geq \Phi$. Then we have $\frac{\delta}{a} \leq \Phi$, or $\delta = O(a\Phi) = O(1)$. This means that the shape can only contract by a constant number of nodes at each step, thus at least $\Theta(\sqrt{k})$ steps to shrink to 1 node. This triangle shape has size $\Theta(k)$.

$\square$