[Reviews · NeurIPS 2014]

Submitted by Assigned_Reviewer_27

This paper addresses the problem of community detection on graphs, via hypothesis testing using a clever idea, characterizing connected subgraphs with linear matrix inequalities to overcome the exponential nature of the problem. Several theorems are stated, in order to justify the proposed statistics, and the method is tested with synthetic data and one real-wold example.

The paper is well organized, but the ideas and theory are squeezed into this 8 pages, making it difficult to follow, and less enjoyable to read. Also, though the organization is correct, the presentation in general (meaning the writing, titles, etc) can be significantly improved.

The addressed problem is classic and of great importance, and the review of the existing work is correct. The approach is new, as far as I know. In particular, I liked the idea of the embedding with linear matrix inequalities, and in fact I would like to see if these ideas apply to other problems related with subgraphs.

Although the paper seems to be a promising idea and technically sound, there are several items missing in the analysis.
For instance, it is not said how the optimization for the statistic is performed, which is a very important part of the method. In particular, the I couldn't find the complexity of the resulting algorithm. How does this method scale with the size of the original graph and with the size of the subgraph?

In my opinion, this paper would be much better without space restrictions, allowing the authors to fully develop the theory and intuition behind the concepts. Also, this paper needs a stronger experimental section, and a rich discussion and conclusion, which are lacking in the present version.
Summary: The paper is strong, specially the theoretical part (the experimental section not so much), but the presentation is the main problem. I think that this paper loses its main strong points in the current format, and would be much better with no space restrictions.

Submitted by Assigned_Reviewer_28

This paper develops a new statistical test to detect anomalies on graphs following the spirit of Generalized Likelihood ratio tests (GLRT). While GLRT requires evaluating a maximum likelihood over a combinatorial class of graphs, the introduced method relaxes this class to a convex set of matrices, such that the maximum expressed as a convex optimization problem under Linear Matrix Inequalities (LMI) constraints. In part 3, the authors show their relaxation explores subsets of connected subgraphs anchored at a given node and introduce their first LMI test (LMIT) for this anchor node and an additional regularization constraint. In part 4, the authors analyze the performance of their test (detectability, separability) and introduce as well an agnostic test which explores the full space of anchor and regularization parameters, which is also proved to achieve asymptotic separability. The performance of LMIT is compared to state of the art approaches in section 5 and is shown to outperform them in terms of area under ROC curve (AUC).
Quality, clarity:
The paper is very well written, easy to follow and with the right level of mathematical complexity. The relevant notions are introduced so that the reader does not need to refer to external sources. The authors also explain intuitively the meaning of the sets of subgraphs that they explore with their approach.
Originality, significance:
As mentioned by the authors, the idea of exploiting LMI constraints in the context of statistical tests on graphs was previously introduced in reference [7]. However, the authors convincingly demonstrate that their new formulation lends itself to a better characterization of the subsets and theoretical performance analysis. The results in theorem 6 and 7 seem impressive since the introduced test is agnostic to shape and size of the graph. The results part are satisfactory, although performance of the agnostic test seems not to be investigated, and comparison of runtime complexity is lacking.
Additional remarks:
It is unclear to me, at least in the main text, how the LMI constrained optimization problem is exploited in practice to evaluate the test statistic. This is an important aspect, since it can also give an idea about the computational complexity of the test.
In the experimental part, it is unclear to me how the anchor is chosen, or whether the test is anchor agnostic. More generally, evaluating the agnostic test seems very heavy computationally, since convex relaxation is applicable only for the non-agnostic case. Does the “practitioner” need to evaluate the test for all possible nodes in the graph as anchor, and all possible shape parameters?

Summary: A good paper introducing a new statistical test on graphs, combining an elegant methodology (using LMI constrained convex relaxation), strong theoretical performance results and some experimental results. Consideration regarding practical implementation and computational time issues are missing.

Submitted by Assigned_Reviewer_42

** I have read the authors' response and the other reviews. The authors have provided satisfactory answers to my concerns. I agree with one of the other reviewers that the presentation is dense, but I think the authors have done a good job considering the space limitations. It would certainly be a better paper with no page limit, but the authors don't have that option. The experimental performance analysis could be improved, but considering the strength of the theoretical contribution, the paper remains for me in the top 15-25 percent of accepted NIPS papers. **

Summary of the paper:

The paper addresses the problem of signal detection for the case where the signals reside on connected subgraphs of a graph that models the relationships between a set of entities. The authors discuss how the problems can be formulated as optimization of objective functions defined on the subgraphs. A straightforward search over the subgraphs is computationally infeasible, so the authors present a highly novel approach that leads to computationally efficient tests. The paper includes proofs that the tests are nearly minimax optimal for the exponential family of distributions and graphs satisfying the polynomial growth property. The paper concludes with an analysis of synthetic and real datasets.

Strengths:

(1) The paper addresses a problem of growing importance and presents novel approaches for statistical tests. These are computationally feasible and the authors have managed to theoretically characterize the performance. The paper is, for the most part, very well-written, so that the concepts and the development of the approach is clear and easy to follow. The proofs are not straightforward and involve techniques that are of interest in their own right.

Weaknesses:

(1) The presented proofs (in the supplementary material) for two of the key results are limited to the 2D-lattice case, but are claimed in the paper for more general settings. The authors argue in each case that this is “For simplicity”, but I don’t think the extension to the claimed polynomial growth is completely obvious, and I’d prefer to see a “For completeness” proof follow the simple arguments. There is also a bold claim that the “analysis extends beyond Poisson & Gaussian models and applies to general graph structures and models” but there is little support for this claim. While the few sentences of argument are reasonable and it is likely that the analysis can be extended as the authors claim, the presented material does not genuinely support the claim.

(2) More detail could be provided for the simulation comparison. Apologies if I missed them, but what are the ground-truth subsets for the lattice examples? Are these constrained to be different from the completely arbitrary connected subsets considered by the SA algorithm? Are they randomized? Why not show a comparison for several choices of \gamma rather than just a value corresponding to thin sets?

Other comments:

The sketch of the proof of Theorem 3 is hard to follow. I think this largely arose for me because I didn’t understand what was meant by “M is connected”. M is a matrix. The only graph associated with it previously in the text is H, so I initially read this as “H is connected”, which means that the argument is circular. After a careful reading of the actual proof, I understand that what is meant by “M is connected” is that the graph with nodes Supp{Diag(M)} and edge set Supp(M) is connected.

In the supplementary material, proof of Theorem 3, I think it is worth adding a sentence prior to (16) explaining that, due to the presence of the anchor node implied by M, the disconnected nature of H can only arise from A, the edge set of the underlying graph. This logical step wasn’t obvious to me at first.

Theorem 3 proof: Line 3: |\overbar{C}| = k_2

The connection between the proof of Theorem 4 and the claim at the end of the proof of Theorem 3 is not immediately obvious. I think it would be better if a direct argument were made in the Theorem 3 proof, or if the authors provided additional clarification regarding which part of the Theorem 4 proof was being applied.

With regard to Theorem 4, it seems odd to me to state a theorem for polynomial growth graphs and then present a proof that (strictly) holds only for 2D-lattices. There isn’t a page limit on the supplementary material and the extension “Similar properties can be shown for polynomial growth graphs,…” is a little flimsy.

A similar comment applies to the proof of Theorem 7; I think it is reasonable to focus on the Gaussian case, because the extension to the Poisson model seems relatively straightforward, but I can’t see a good reason to include only a sketch of the proof for the 2D-lattice when claiming a general result.
Summary: The paper addresses the problem of signal detection for the case where the signals reside on connected subgraphs of a graph that models the relationships between a set of entities. This is a problem of growing interest and important; the presented statistical tests are novel and computationally feasible and the authors have managed to theoretically characterize the performance. The paper is very well-written, so that the concepts and the development of the approach is clear and easy to follow. The proofs are not straightforward and involve techniques that are of interest in their own right.
Author Feedback
Author rebuttal: +Rev 1

Experiment section
First, we only found one relevant real-world dataset (disease outbreak). There are innumerable applications but datasets are unavailable. Most related work except [7] construct synthetic datasets. [7] has the same dataset as ours.
So our goal was to demonstrate the impact of shape & graph sparsity on detection performance to verify our theoretical results. We compare against other state-of-art approaches on lattice, RGG and on the one real-world dataset to demonstrate various issues that we are unable to analyze such as finite graphs etc. We feel that the experiments convey useful information about our approach given the current state-of-art as well as provides directions for future research.

+Rev 1&2

A) Test statistic Clarification.
We first clarify our algorithm. Given signal x, we fix an anchor & shape parameter and solve: Max: diag(M)’x s.t. constraints of (9). We then compute diag(M)’x/sqrt( tr(M) ). Then we iterate over the different anchors to find the maximum value. The test is then performed by thresholding. In the purely agnostic case we also maximize over gamma before thresholding.

B) Complexity:
The problem is a linear cost problem with LMI constraints plus other linear constraints. The size of the LMI is n \times n where n is the number of nodes in the graph. The number of linear constraints are about m ~ number of edges. The number of unknowns in general (dense graph) is about O(n^2).
However, for sparse graphs the number of unknowns can be reduced to O(m), i.e., variables corresponding to nodes and edges.

The problem has polynomial sample complexity in terms of n & m for a fixed anchor and fixed gamma. Iterating over the anchor adds an another n to the sample complexity. gamma is scalar and bounded variable that we can quantize and search.

We use the built-in CVX/SeDuMi solver of Matlab to solve this LMI constrained problem. For dense graphs with n nodes and O(n^2) edges, these solvers can only deal with n ~ 300 nodes. For sparse graphs like lattice, SeDuMI can scale up to n ~ 2000 nodes. We observe that the main reason for the relatively small n is that memory becomes an issue. In general solving such problems is currently an active line of research which we plan to leverage in the future.

+Rev 2

C) Agnostic issue
If completely agnostic to anchor and shape, the test needs to be performed for different anchors / shape parameters as in Eq.(12,13). See also (A)

+Rev 3

D) Ground-truth for Table 1

We apologize for the missing figures of lattice & RGG for Table 1, which we thought had been presented at the beginning of the supplementary material but inadvertently got left out (perhaps loaded the wrong file). Just to clarify we fix the ground-truth shape and vary Gaussian noise for different runs.

E) “M is connected” issue
We agree. Perhaps the following sentence could be added after Defn.7 to clarify this issue. From now on “M is connected” means that M corresponds to H by Defn.7 and H is connected.

F) Connection between proof of Thm.3 & Thm.4

We agree. This point has to be clarified further and will plan to do so. Basically, we want to allow for choice of shapes if this is known a priori. Thm 3 is simply suggesting to ensure connectivity all we need to do is to choose gamma>0. Thm 4 is suggesting that different values of gamma result in different shapes.

G) Proofs for 2D lattice & extension to polynomial growth graphs

We agree. We plan to re-state several statements here to clarify cases being proved and provide more intuition.